# Antibacterial and Freshness-Preserving Mechanisms of Chitosan-Nano-TiO₂-Nano-Ag Composite Materials

**Zihao Dong** [1,2,†], **Ran Li** [1,†] and **Yan Gong** [1,2,*]

1   School of Life Sciences, Shanxi Normal University, Linfen 041004, China; dongzihao_97@163.com (Z.D.); liRan4071433@163.com (R.L.)
2   Modern College of Arts and Sciences, Shanxi Normal University, Linfen 041004, China
*   Correspondence: crush_bc@163.com
†   These authors contributed equally to this work.

**Abstract:** With chitosan, nano-TiO₂ and nano-Ag as raw materials, nano-TiO₂ and nano-TiO₂-Ag were modified by a surface modifier-sodium laurate. Chitosan (CTS), chitosan-nano-TiO₂ (CTS-TiO₂), and chitosan-nano-TiO₂-nano-Ag (CTS-TiO₂-Ag) composite materials and corresponding films were prepared by a solution co-blending method. Then, the antibacterial performances of the above three types of materials against *Escherichia coli*, *Staphylococcus aureus*, and *Bacillus subtilis* were compared. Moreover, potato and strawberry weight loss rates, peroxidase activity, and vitamin C contents after different film coating treatments were measured. Compared with CTS films, the CTS-TiO₂-Ag and CTS-TiO₂ composite films both showed better physical properties, and both demonstrated higher antibacterial effects, especially for *E. coli*. Measurement of physiological indices in fruits and vegetables showed that the freshness-preserving effect of CTS-TiO₂-Ag coating films was the most significant. In all, the CTS-TiO₂-Ag coating films can actively contribute to the storage of fruits and vegetables at room temperature, and better ensure product quality. Thus, such films are meaningful for research and development of new fruit freshness-keeping techniques and materials.

**Keywords:** nano-TiO₂; nano-TiO₂-Ag; CTS-TiO₂-Ag; antibacterial performance; freshness preservation





## 1. Introduction

With the continual rise of market demands for fruits and vegetables in recent years, the rotting and deterioration in the transportation of fruits and vegetables from mountainous areas to cities become major problems that trouble farmers and dealers. In China, the loss rates of fruits, vegetables, and other agricultural subsidiary products amid picking, transportation, storage, and other circulation steps are up to 30%. Particularly, 0.037 billion tons of fruits and vegetables are rotten on the road every year, and the direct annual economic losses of fruits and vegetables in China are up to 100 billion Yuan. Traditional freshness-keeping techniques cannot satisfy the transporting requirements of fruits and vegetables [1,2], and will cause environmental pollution and other problems that harm human health [3,4]. To decrease the waste of fruits and vegetables and improve freshness and quality, the need for finding new freshness-keeping techniques is very urgent.

Chitosan, or (1,4)-2-amino-2-deoxygen-β-D-glucosan [5], results from partial or complete deacetylation of crude polysaccharide chitin [6]. The amino groups in its molecular structure are more reactive than the acetylamino group of chitin, and therefore chitosan has many excellent biological functions, such as high biological compatability, nontoxicity, antibacterial capability, and film-forming ability [7,8]. Zhao et al. used antibacterial and reducible chitosan composites as antiseptics on strawberries and litchi, and systematically screened CTS-CS₂-Zn and CTS-CS₂-Ag with special antibacterial capabilities against *Bacillus subtilis* and *Staphylococcus aureus*, respectively [9]. Ding et al. synthesized RS-CTS from chitosan and salicylaldehyde and confirmed its antibacterial effects by using ultraviolet spectrometry and infrared spectroscopy [10]. Edible coatings can potentially decrease

losses of gardening products after picking. A study on herbage fragrant plants, including basil, demonstrated that chitosan and chitosan nanoparticles used as edible coatings can largely prolong the shelf life of basil leaves and prevent severe losses [11]. Valenzuela et al. found that chitosan-based edible coatings can solve strawberry rotting [12].

Matsunaga et al. from Japan first reported the antibacterial effects of $TiO_2$ photocatalytic reaction in 1985 [13]. After that, $TiO_2$ photocatalysis was further studied to kill viruses [14], bacteria, fungi, algae, and cancer cells [15,16]. Nanoscale $TiO_2$ antibacterial agents are more effective and can destroy cell walls and membranes to decompose microbes, showing the microbe-inhibiting or even microbe-killing effects, and can decompose the ethylene released during fruit and vegetable storage into water and $CO_2$ [17], thus prolonging the storage time [18]. Cho et al. used nano-$TiO_2$/UV photocatalysts to disinfect surface bacteria on carrots and found the shelf life of processed carrots was prolonged [19]. Long et al. tested the antibacterial effects of nano-$TiO_2$ on Nanfeng citrus during storage, and found nano-$TiO_2$ can inhibit penicillia, *Penicillium digitatum*, and *Escherichia coli* parasitizing citrus, showing significant freshness-keeping effects [20].

Nano-Ag refers to the silver elementary substance with at least one of three-dimensions [21] at the nanometer scale (particle size < 100 nm) [22,23]. Besides outstanding heat-resistance, light resistance, and chemical stability, nano-Ag powder can accelerate ethylene oxidation and release from fruits and vegetables, decreasing ethylene concentrations in packages, efficiently prolonging antibacterial time [21,24], and improving freshness and quality [25,26]. The Ag-$TiO_2$ composite materials synthesized from the inhibition zone method can inhibit some Gram-negative bacteria (e.g., *E. coli*) and Gram-positive bacteria (e.g., *S. aureus*), indicating the strong antibacterial activity of Ag-$TiO_2$ [23]. Xue et al. synthesized Ag nanoparticles in situ that can improve the antibacterial ability of chitosan [27]. A study targeting the antibacterial effects of coatings with or without nano-Ag particles on bacili shows that coatings loaded with nano-Ag particles are more practical, with an antibacterial rate up to 91.9% [28].

So far, synthesis of composite films from nano-$TiO_2$ and edible chitosan is increasingly studied. The antibacterial effects of composite films with chitosan as the substrate and added with nano-$TiO_2$ are better. However, there is little research on preparing composite films from nano-Ag, nano-$TiO_2$, and edible chitosan for fresh preservation of fruits and vegetables. Hence, comprehensive and profound research on antibacterial performances and fresh preservation of chitosan-nano-$TiO_2$-nano-Ag is very necessary. In this study, nano-Ag and nano-$TiO_2$ were mixed, and nano-$TiO_2$ and $TiO_2$-Ag were modified by sodium laurate. The modified materials were characterized by X-ray photoelectron spectroscopy (XPS), scanning electron microscopy (SEM), and Fourier transform infrared spectroscopy (FTIR). CTS films, CTS-$TiO_2$ composite films, and CTS-$TiO_2$-Ag composite films were prepared from a solution co-blending method and then their reflectance, transmissivity, and absorbance values were measured. The inhibitory effects of CTS, CTS-$TiO_2$, and CTS-$TiO_2$-Ag against *E. coli*, *S. aureus*, and *B. subtilis* were compared using an Oxford cup method. Fresh preservation experiments of potato and strawberry were conducted at room temperature. Changes in weight loss rates, peroxidase activity, and vitamin C (VC) concentrations of potato and strawberry during storage were detected. The quality and physiologic indices of potatoes and strawberries during storage were experimentally studied. The fresh-keeping effects of CTS, CTS-$TiO_2$, and CTS-$TiO_2$-Ag coated films were investigated, aiming to underlie the application of nanotechnology into fruits and vegetables anti-corrosion and fresh preservation.

## 2. Materials and Methods

### 2.1. Main Materials and Instruments

Reagents used here included nano-$TiO_2$, nano-Ag powder, chitosan and sodium laurate (Shanghai Aladdin Bio-Chem Technology Co., Ltd., Shanghai, China), $KH_2PO_4$ (Tianjin Kemiou Chemical Reagent Co., Tianjin, China), $K_2HPO_4$ (Luoyang Chemical Reagent Plant, Luoyang, China), $NaH_2PO_4$ (Tianjin Guangfu Techonology Co. Ltd., Tianjin,

China), and ice acetic acid (Tianjin Shentai Chemical Reagent Co., Ltd., Tianjin, China). *E. coli*, *S. aureus*, and *B. subtilis* were bought from ICloning Beijing Biotech Co., Ltd., Beijing, China. Petri dishes (100 mm × 100 mm) were purchased from Membrane Solutions Co., Ltd., Beijing, China. Relevant instruments included an XPS device (USA Thermo escalab 250Xi, Cambridge, MA, USA), an infrared spectrometer (both Thermo Nicolet Corporation, Madison, WI, USA), a field-emission SEM device (Hitachi, Tokyo, Japan), an ultraviolet-visible pectrophotometer (Shanghai Metash Instruments Co., Ltd., Shanghai, China), magnetic heated stirrers (Changzhou Jintan Earth Automation Instrument Factory, Changzhou, China), vacuum drying ovens (Shanghai Boxun Industry & Commerce Co., Ltd., Shanghai, China), and electronic balances (Shanghai Zhuojing Electronic Technology Co., Ltd., Shanghai, China).

*2.2. Preparation of Nano-Ag-TiO$_2$ Composite Materials*

Nano-TiO$_2$ (0.7 g) was added into 35 mL of deionized water and ultrasonically stirred at 50 KW for 3 min (SK5210IIP, Kedao Ultrasonic Instrument Co., Ltd., Shanghai, China). Nano-Ag powder (0.118 g) was dissolved into the above solution and ultrasonically stirred for 15 min, forming a uniform solution [29].

*2.3. Modification*

2.3.1. Modification of Nano-TiO$_2$

Nano-TiO$_2$ (0.7 g) was added into 35 mL of deionized water and ultrasonically pre-dispersed. Then, 0.105 g of a sodium laurate solution was added [30] and ultrasonically dispersed. After adjustment to pH 5, reaction at 40 °C proceeded for 30 min. The resulting products were filtered, washed with toluene four times, and dried, forming white nano-TiO$_2$ with surface lipophilicity.

2.3.2. Modification of Nano-Ag-TiO$_2$

Into the nano-Ag-TiO$_2$ uniform solution above, 0.1227 g of a sodium laurate solution was added and ultrasonically dispersed. After adjustment to pH 5, reaction at 40 °C proceeded for 30 min. The resulting products were filtered, washed four times with toluene, and dried, forming nano-Ag-TiO$_2$ with surface lipophilicity.

*2.4. Preparation of Films*

2.4.1. Preparation of Chitosan Films

Chitosan (2.0 g) was dissolved into 100 mL of 0.6% ice acetic acid solution (volume fraction) and stirred by magnetic stirrers for complete dissolution. After ultrasonic degassing for 15 min, the solution was stirred and degassed for three repetitions, forming a chitosan film solution. The chitosan film solution was ultrasonically degassed for 10 min and quantitatively coated onto horizontal glass plates, so the solution was flow-cast into films, which were dried under an infrared lamp. The dried films were soaked in a 1.0 mol/L NaOH solution for 30 min and neutralized by washing with abundant flow water. Then the films were carefully taken off and dried at room temperature. The single films were stored in a desiccator and marked as CTS.

2.4.2. Preparation of Chitosan-Nano-TiO$_2$ Composite Films

The activated nano-TiO$_2$ (0.035 g) and glycerol (2 g) were added into 100 mL of a glacial acetic acid solution with a volume fraction of 0.6%. Then, 2 g of chitosan were added and stirred by magnetic stirrers for complete dissolution. The resulting solution was ultrasonically degassed for 15 min, stirred and degassed for three repetitions. The resulting CTS-TiO$_2$ composite film solution was flow-cast, dried, washed, and taken off as per Section 2.4.1. The final composite film solution was marked CTS-TiO$_2$.

### 2.4.3. Preparation of Nano-Ag-TiO$_2$-Chitosan Composite Films

Activated nano-TiO$_2$ (0.035 g) was dissolved into 2 g of glycerol and added with 100 mL of a 0.6% ice acetic acid solution (volume fraction). Then, 2 g of chitosan was added and stirred by magnetic stirrers for complete dissolution. The resulting solution was ultrasonically degassed for 15 min, stirred and degassed for three repetitions. The resulting CTS-TiO$_2$-Ag composite film solution was flow-cast, dried, washed, and taken off as per Section 2.4.1. The final composite film solution was marked as CTS-TiO$_2$-Ag.

### 2.5. Characterization of Samples

#### 2.5.1. Characterization of Composite Materials

(1) The compositions of CTS-TiO$_2$ composites and CTS-Ag-TiO$_2$ composites were observed by the Thermo ESCALAB 250Xi XPS device (Thermo, Cambridge, MA, USA) [31].
(2) Surface morphology was observed under the TecninaiG2F30S-TWIN SEM device (Hitachi, Tokyo, Japan) [30].

#### 2.5.2. Characterization of Composite Films

Transmissivity, reflectance, and absorption of the three types of films were measured by the UV-3600 UV spectrophotometer (Shanghai Metash Instruments Co., Ltd., Shanghai, China) [32].

### 2.6. Antibacterial Performances

Antibacterial experiments of the samples against *E. coli*, *B. subtilis*, and *S. aureus* were tested by an Oxford cup method [33], and the sizes of inhibition zones after certain periods of time were recorded.

(1) Preparation of culture medium

An appropriate amount of agar powder was added into a Luria–Bertani (LB) bacterial culture medium, which was sterilized in a high-pressure sterilization pot for 120 °C. Then the medium was taken out and cooled on an aseptic operation platform to 60 °C. Then each culture dish was added with an appropriate amount of the medium until the whole bottom was covered. After cooling and solidification, *E. coli* was inoculated [34].

(2) Procedures of antibacterial experiments

All glassware and materials were autoclaved at 120 °C for 15 min to ensure sterility of the tests. The tested bacterial suspensions at concentrations of $10^7$–$10^8$ cfu/mL were dipped with an inoculation ring and uniformly applied three times, each time 60, to the surface of nutrient agar medium plates. The plates were covered and dried at room temperature for 5 min. Then, sterilized Oxford cups were placed onto the three coated culture dishes, which were added with CTS, CTS-TiO$_2$, and CTS-TiO$_2$-Ag, respectively. In each group, the Oxford cups were uniformly distributed at three directions of culture dishes. The three types of antibacterial culture dishes were put into a thermostatic incubator for 24 h of cultivation at 37 °C. Then, the inhibition zone diameter was measured, and the antibacterial ability of each type of film was evaluated.

The suspensions of the three test bacteria were $10^7$–$10^8$ cfu/mL. Experiments with *B. subtilis* and *S. aureus* were conducted in the same way.

### 2.7. Measurement and Methods of Indices of Fruits and Vegetables

#### 2.7.1. Sample Processing

Mature potatoes in uniform size and without mechanical damage or plant diseases and insect pests were selected and cut into pieces, which were randomly divided into four groups (each 20 pieces). Then, the potatoes were coated with films. The pieces were soaked in corresponding solutions for 2 min and then taken out. After natural air drying, the potatoes were sent to CTS, CTS-TiO$_2$, or CTS-TiO$_2$-Ag film fresh-keeping experiments at room temperature, with a blank control. During the storage, all indices were measured once every 2 days, and three pieces at each time were randomly chosen from each group.

Mature strawberries in uniform size and without mechanical damage or plant diseases and insect pests were selected and randomly divided into four groups (each 20 strawberries). Then, the strawberries were coated with films. The strawberries were soaked in corresponding solutions for 2 min and then taken out. After natural air drying, the strawberries were sent to CTS, CTS-TiO$_2$, or CTS-TiO$_2$-Ag film fresh-keeping experiments at room temperature, with a blank control. During the storage, all indices were measured once every 1 day, and three strawberries at each time were randomly chosen from each group.

### 2.7.2. Items and Methods of Measurement

The weight loss rate was measured by a weight method and calculated as follows: weight loss rate (%) = ((weight before storage − weight after storage)/weight before storage) × 100%. VC concentration was measured by UV spectrophotometry [35]. Peroxidase (POD) activity was detected by guaiacol colorimetry [36,37].

### *2.8. Data Processing*

Experimental data were statistically analyzed on Excel and SPSS, and plotted on Origin. Each processing was repeated at least three times, and data were expressed as mean ± standard deviation of three independent experiments. Significant differences of Student *t*-test were set at * $p < 0.05$, ** $p < 0.01$, and *** $p < 0.001$.

### 3. Results and Discussion
### *3.1. SEM, XPS and FT-IR Characterization of TiO$_2$-Ag*

Figure 1a,b shows the SEM images of TiO$_2$-Ag [38], from which the morphology and sizes of particles can be observed, where the majority of particle sizes is around 100 and 500 nm, respectively. The particles are uniform and obviously aggregate. The average size of nanoparticles is 157 nm × 273 nm. The surface of nanoparticles shows numerous dislocations, stacking faults and other defects, and increasing the surface energy and promoting mutual binding and aggregation. Clearly, the surface of TiO$_2$ was successfully coated with Ag, which is similar to a previous report [39]. The Ag nanoparticles mainly adhere to TiO$_2$, and uniform distribution of TiO$_2$ also promotes and improves the broad-spectrum antibacterial abilities of Ag nanoparticles.

Figure 1c demonstrates the XPS spectrum of TiO$_2$-Ag. The complete spectral measurement reveals peaks of TiO$_2$-Ag at C1s (285.08 eV), Ag3d (367.68 eV), Ti2p (458.88 eV), and O1s (530.08 eV). C1s is ascribed to the residual carbon of the precursor, which is consistent with some experimental results reported before [40]. The peak value at Ag3d is consistent with the standard spectrum, which proves that Ag exists as elementary substance and indicates Ag was mixed in the form of elementary substance into TiO$_2$. The peak of Ti2p at 458.88 eV corresponds to the electron-binding energy of Ti2p3/2 [41]. The peak of O1s at 530.08 eV is ascribed to the electron-binding energy of crystal lattice oxygen (Ti-O). XPS confirms the successful preparation of TiO$_2$-Ag composite structures.

The surface functional groups of TiO$_2$ and TiO$_2$-Ag were detected by FT-IR (Figure 1d). Clearly, the FT-IR spectra of the two samples show similar trends, and the absorption peaks are located very close, indicating the presence of similar surface chemical groups. The two characteristic absorption peaks of TiO$_2$-Ag at 2919.31 and 2850.56 cm$^{-1}$ originate from the stretching vibration of the associated carboxyl group in C-H on the surface functional groups of TiO$_2$-Ag. The peak at 1713.26 cm$^{-1}$ is attributed to the stretching vibration of C=O in the carboxyl group. The peak at 1467.15 cm$^{-1}$ reflects the bending vibration of the methyl group. The sharp peak at 449.06 cm$^{-1}$ is caused by the deformation vibration of C–C–C. The infrared spectra show that nanomaterials are rich in the carboxyl group and the hydroxyl group, which decide the ideal water solubility of nanomaterials. Since the surface of TiO$_2$-Ag contains C=O and C–C–C, Ag$^+$ can combine with oxygen atoms in these groups to form stable coordination compounds. Hence, this is the theoretical basis of modifying TiO$_2$ by doping Ag.

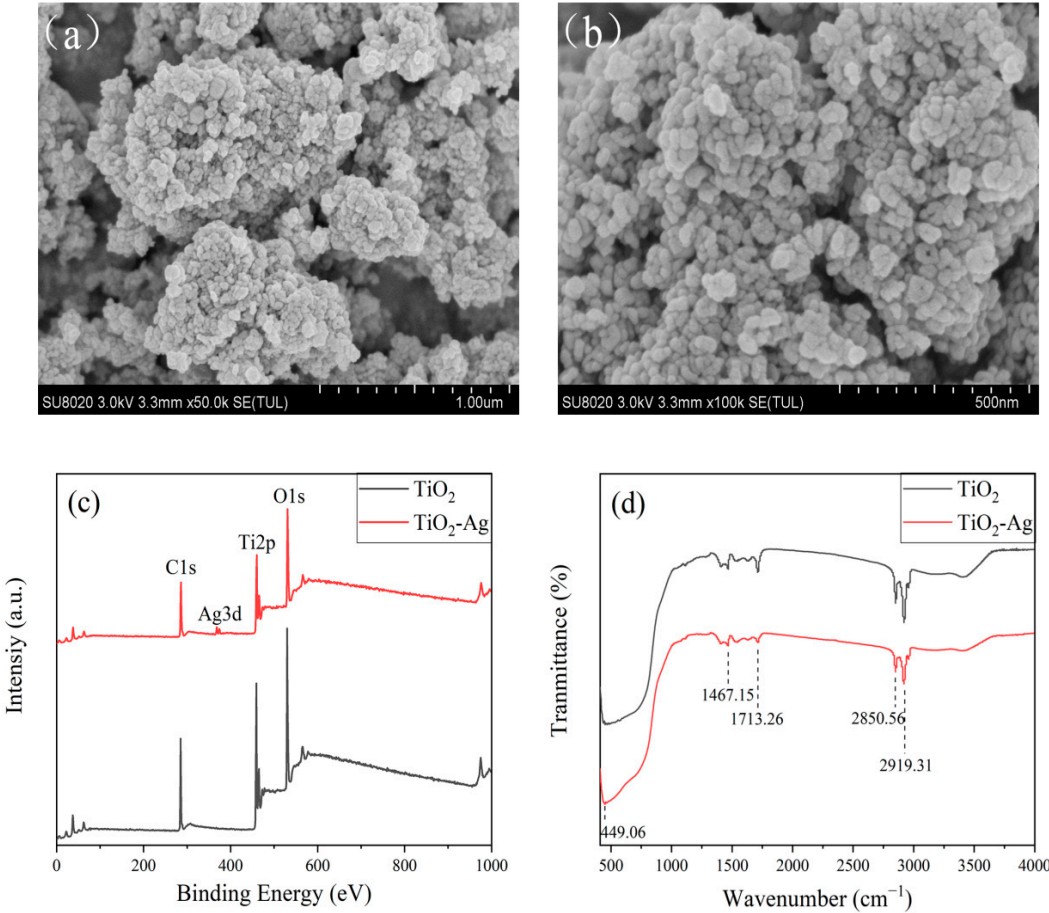

**Figure 1.** Characterization analysis diagram of TiO$_2$-Ag: (**a**) SEM (1 μm) images of TiO$_2$-Ag; (**b**) SEM (500 nm) images of TiO$_2$-Ag; (**c**) XPS spectra of TiO$_2$-Ag; (**d**) FT-IR spectrum of TiO$_2$-Ag.

### 3.2. Comparison of Chitosan Film, Nano-TiO$_2$-Chitosan Composite Film, and Nano-Ag-TiO$_2$-Chitosan Composite Films

### 3.2.1. Reflectivity

Reflectivity refers to the percentage of reflected radiative energy accounting for the total radiative energy of an object, and higher reflectivity indicates the absorption of less solar radiation. As for films, a film with larger reflectivity absorbs less solar radiation, and the fruits and vegetables covered by this film will absorb less solar irradiation, so the rotting of fruits and vegetables is gentle. From 200–350 nm to the UV region, UV has a sterilization function, and CTS-TiO$_2$-Ag and CTS-TiO$_2$ have higher reflectivity compared with CTS. In the visible region, from 400–800 nm, the reflectivity of CTS-TiO$_2$, or CTS-TiO$_2$-Ag films is higher than that of CTS films, and that of CTS-TiO$_2$-Ag composite films is higher (Figure 2). These results indicate that, under the same conditions, the fruits and vegetables covered by CTS-TiO$_2$-Ag composite films absorb less solar radiation and are not prone to oxidation or moisture loss, thus showing strong freshness-keeping ability.

### 3.2.2. Transmissivity

Chitosan films are transparent, however, CTS-TiO$_2$ composite films and CTS-TiO$_2$-Ag composite films are non-transparent. The light transmittance of the three types of films varies to different extents at 200 to 400 nm (Figure 3), and the transmissivity rates of CTS-TiO$_2$ composite films and CTS-TiO$_2$-Ag composite films are significantly lower than that of CTS films. These results indicate that the addition of nano-TiO$_2$ may alter the infiltration route of water molecules in the films and enhance the water resistance of films, increasing the moisture preserving ability [23].

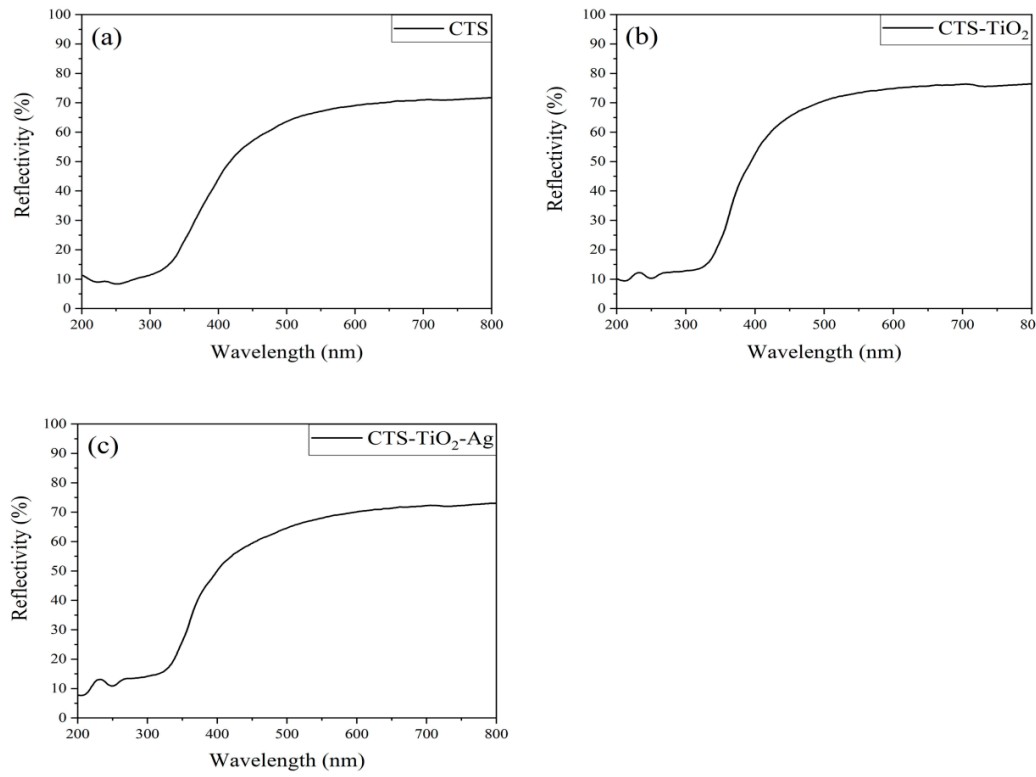

**Figure 2.** The reflectivity of the three films: (**a**) CTS single membrane; (**b**) CTS-TiO$_2$ composite membrane; (**c**) CTS-TiO$_2$-Ag composite membrane.

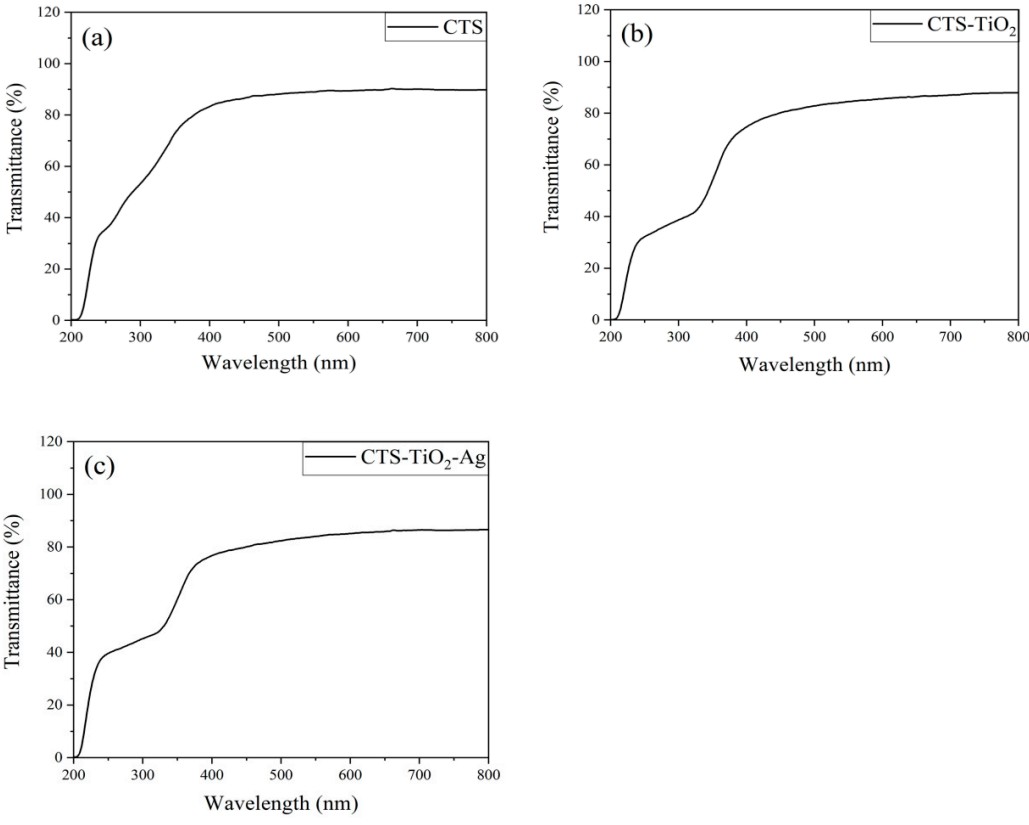

**Figure 3.** The transmittance of three films: (**a**) CTS single membrane; (**b**) CTS-TiO$_2$ composite membrane; (**c**) CTS-TiO$_2$-Ag composite membrane.

### 3.2.3. Absorption Value

Absorption reflects the light absorbing ability of substances. The absorption values of CTS, CTS-TiO$_2$, and CTS-TiO$_2$-Ag composite films rapidly decline within 200–250 nm (Figure 4). Within 250–350 nm, the absorption value of CTS films is still declining at the same rate, however, the values of CTS-TiO$_2$ and CTS-TiO$_2$-Ag composite films drop first slowly and then quickly. The absorption values at 350 nm rank as CTS-TiO$_2$-Ag composite films > CTS-TiO$_2$ composite films > CTS films. Some scientists have proved that the silver additive enhanced the absorbance of the film in the visible light region [42].

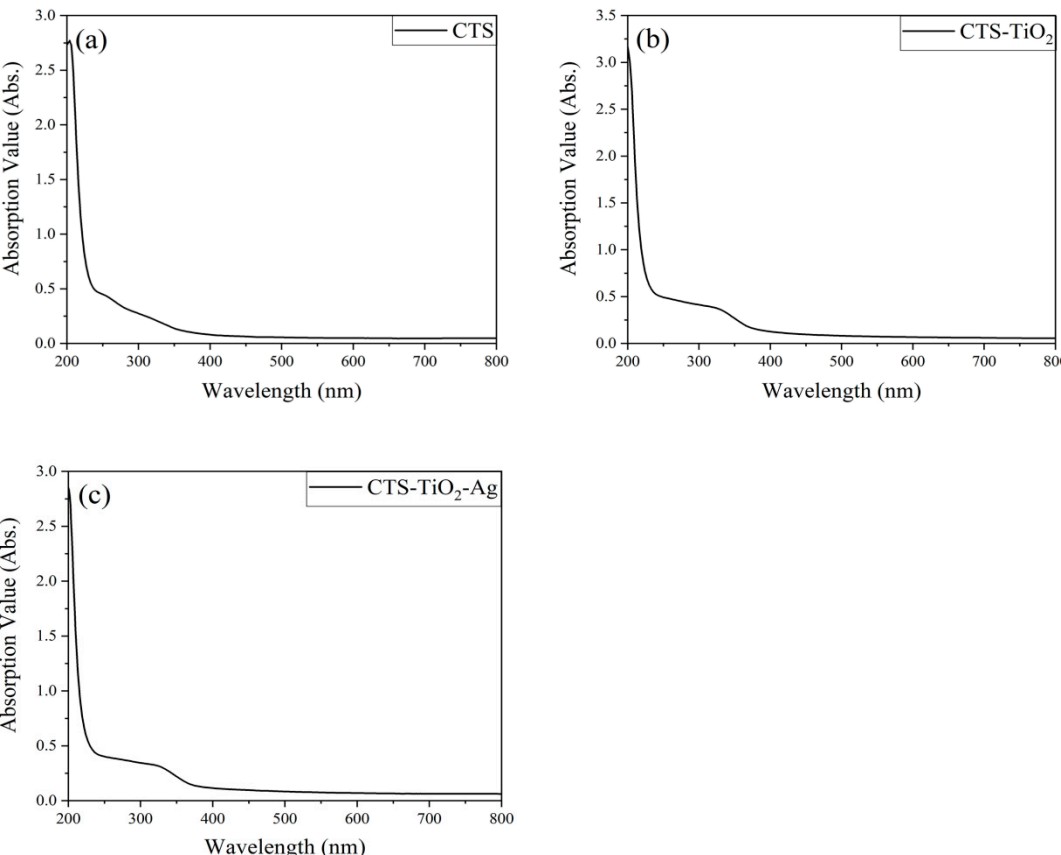

**Figure 4.** The absorption values of the three films: (**a**) CTS single membrane; (**b**) CTS-TiO$_2$ composite membrane; (**c**) CTS-TiO$_2$-Ag composite membrane.

### 3.3. Antibacterial Performance of Composite Materials

The three types of films yielded antibacterial rings in different sizes on the three types of plates (Figure 5), indicating the antibacterial effects of the antibacterial films differ depending on the type of bacteria. In addition, Table 1 more clearly shows the differences in the antibacterial effects among the three different materials. The inhibition results of the three materials on *E. coli* show that the inhibition zone diameters of CTS-TiO$_2$-Ag, CTS-TiO$_2$, and CTS are 22.00 mm > 21.83 mm > 16.33 mm, respectively. The inhibition zone diameters of CTS-TiO$_2$-Ag, CTS-TiO$_2$, and CTS against *S. aureus* are 20.33 mm > 20.17 mm > 16.17 mm, respectively. For *B. subtilis*, only an obvious inhibition zone (20 mm) was found at CTS-TiO$_2$-Ag, and the inhibitory effects of CTS-TiO$_2$ and CTS on *B. subtilis* were not obvious. Moreover, the antibacterial strength of the three materials ranks CTS-TiO$_2$-Ag > CTS-TiO$_2$ > CTS. The inhibition results of the same material against different bacteria are as follows: the inhibition zone diameters of the materials against *E. coli*, *S. aureus*, and *B. subtilis* are 16.33 mm > 16.17 mm > 0 (CTS); 21.83 mm > 20.17 mm > 0 (CTS-TiO$_2$); 22.00 mm > 20.33 mm > 20 mm (CTS-TiO$_2$-Ag). Clearly, the same material has the strongest inhibitory effect on *E. coli*, followed by *S. aureus* and

*B. subtilis*. Based on the above experiments, the antibacterial spectra of CTS, CTS-TiO$_2$, and CTS-TiO$_2$-Ag were clarified, which showed that the antibacterial spectrum of CTS-TiO$_2$-Ag is the broadest, and this composite material also has a strong inhibiting ability toward *E. coli* and *S. aureus* [24]. In addition, research by scientist has demonstrated that that film carrying the silver nanoparticles exhibits enhanced antibacterial activity against *E. coli* and *S. aureus* [43].

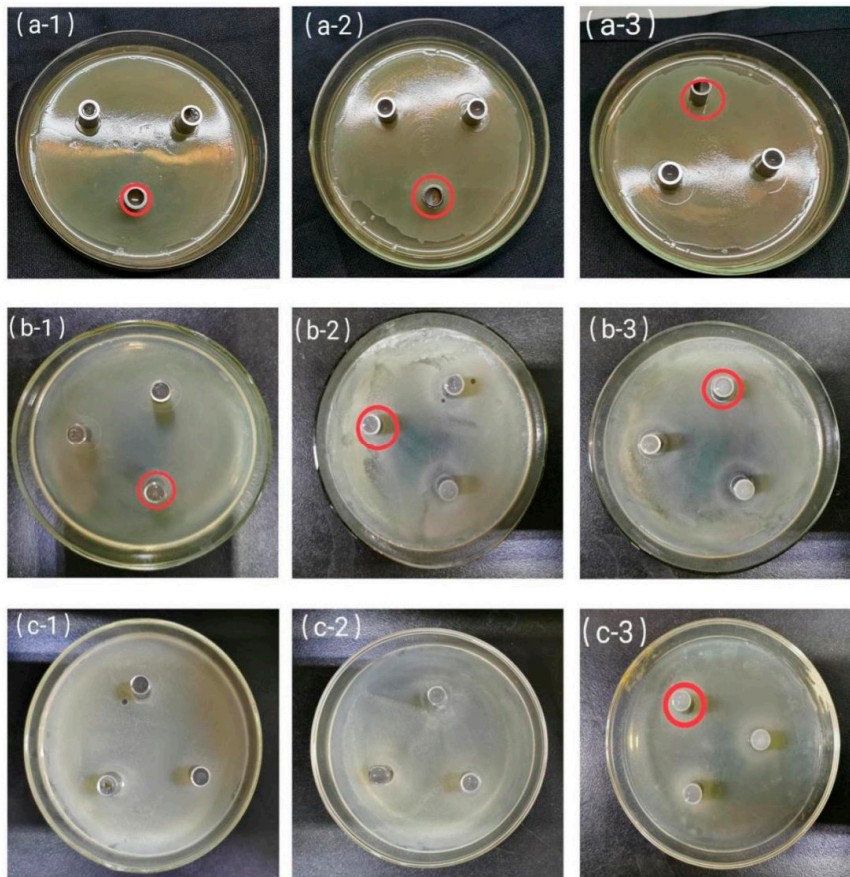

**Figure 5.** Oxford cup test (1) CTS; (2) CTS-TiO$_2$; (3) CTS-TiO$_2$-Ag. Composite material pair: (**a**) Escherichia coli; (**b**) Staphylococcus aureus; (**c**) Antibacterial property of bacillus subtilis.

**Table 1.** Inhibition zone size of CTS, CTS-TiO$_2$ and CTS-TiO$_2$-Ag after treatment.

| Treatment | Size of Inhibition Zone (mm) |
| --- | --- |
| Escherichia coli-CTS | $16.33 \pm 0.76$ *** |
| Escherichia coli-CTS-TiO$_2$ | $21.83 \pm 0.58$ *** |
| Escherichia coli-CTS-TiO$_2$-Ag | $22.00 \pm 0.87$ *** |
| Staphylococcus aureus-CTS | $16.17 \pm 1.04$ *** |
| Staphylococcus aureus-CTS-TiO$_2$ | $20.17 \pm 1.04$ *** |
| Staphylococcus aureus-CTS-TiO$_2$-Ag | $20.33 \pm 0.76$ *** |
| Antibacterial property of bacillus subtilis-CTS | 0 |
| Antibacterial property of bacillus subtilis-CTS-TiO$_2$ | 0 |
| Antibacterial property of bacillus subtilis-CTS-TiO$_2$-Ag | $20.00 \pm 0.50$ *** |

Data given demonstrate the average of three replications (n = 3) $\pm$ SD. Asterisks within rows indicate significant differences (* $p < 0.05$, ** $p < 0.01$, *** $p < 0.001$), according to Student's *t*-test.

### 3.4. Fresh Preservation of Fruits and Vegetables by Composite Films

3.4.1. Weight Loss Rate

Weight loss rate is one major indicator of the freshness of fruits and vegetables [44]. Fruits and vegetables during storage will respire and evaporate, causing moisture dissipation, so the weight loss rates of fruits and vegetables will rise with the prolonging of storage. The weight loss rates of potatoes during storage are illustrated in Figure 6a. During the storage, weight loss occurred in all groups of potatoes. After 8 days of storage, the average weight loss rates were CK: 9.485%, CTS: 8.52%, CTS-TiO$_2$: 7.09%, and CTS-TiO$_2$-Ag: 6.62%. The weight loss rates of the experimental groups were all smaller to that of the control group, and minimized in the CTS-TiO$_2$-Ag group. The weight loss rates of strawberries in the experimental groups are shown in Figure 6b, and weight loss occurred in all groups of strawberries during the storage. Water loss of strawberries during storage was very severe in the control group and the CTS group. After six days, the weight loss rate of the unpacked strawberries suddenly increased severely. This was because strawberry rotting was intensified on this day. After eight days of storage, the weight loss rates of the experimental groups and control group were CK: 11.71%, CTS: 10.56%, CTS-TiO$_2$: 8.06%, CTS-TiO$_2$-Ag: 7.47%, which exhibit the same trend as in the experiments on potatoes. Reportedly, the coating of fruits with films can reduce respiration and consequently lead to a decrease in ethylene production and loss of mass, which contribute to a longer conservation period [45]. The addition of AgNPs in coating films is considered an alternative to improve the mechanical, thermal, gas, and water barriers of the films [46].

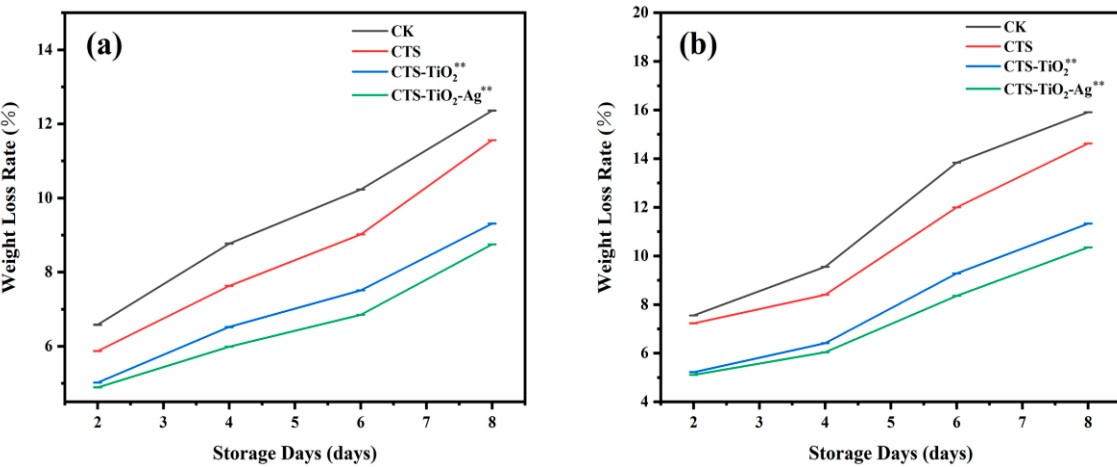

**Figure 6.** Effect of treatment with three different materials on weight loss rate: (**a**) potato; (**b**) strawberry. CK: Potatoes and strawberries without any treatment; CTS: CTS applied to form film-treated potatoes and strawberries; CTS-TiO$_2$: CTS-TiO$_2$ applied to form film-treated potatoes and strawberries; CTS-TiO$_2$-Ag: CTS-TiO$_2$-Ag applied to form film-treated potatoes and strawberries. Data given demonstrate the average of at least three replications (n = 3) $\pm$ SD. Asterisks within rows indicate significant differences (* $p < 0.05$, ** $p < 0.01$, *** $p < 0.001$), according to Student's $t$-test.

3.4.2. Measurement of Peroxidase Activity

Peroxidase is ubiquitous in plants and is an active and adaptive enzyme in the antioxidase system. Peroxidase activity effectively reflects the internal metabolic condition, growth, development, and environmental adaptability of plants [47]. Compared with the blank control (CK), the peroxidase activities of CTS-TiO$_2$-Ag, CTS-TiO$_2$, and CTS groups are 11.67%, 7.95%, and 3.71% higher, respectively (potatoes), and are 14.27%, 9.9%, and 3.70% higher, respectively (strawberry) (Figure 7). These results indicate nanomaterials coated on the surface of potatoes and strawberries can efficiently enhance peroxidase activity, antioxidant capacity, and food freshness, and CTS-TiO$_2$-Ag is the most effective in enhancing peroxidase activity.

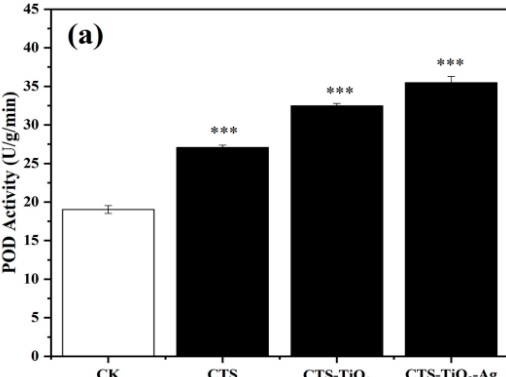
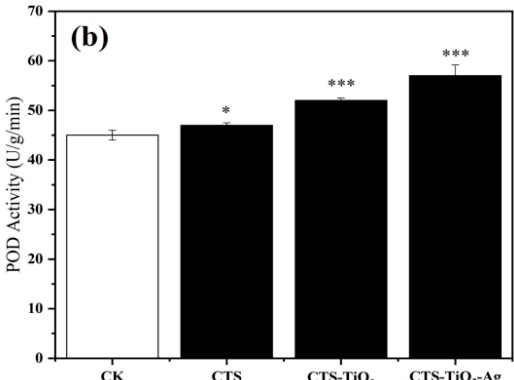

**Figure 7.** Effect of treatment with three different materials on peroxidase activity: (**a**) potato; (**b**) strawberry. CK: Potatoes and strawberries without any treatment; CTS: CTS applied to form film-treated potatoes and strawberries; CTS-TiO$_2$: CTS-TiO$_2$ applied to form film-treated potatoes and strawberries; CTS-TiO$_2$-Ag: CTS-TiO$_2$-Ag applied to form film-treated potatoes and strawberries. Data given demonstrate the average of at least three replications (n = 10) ± SD. Asterisks within rows indicate significant differences (* $p < 0.05$, ** $p < 0.01$, *** $p < 0.001$), according to Student's *t*-test.

### 3.4.3. Measurement of VC Concentration

VC concentration is a major indicator of nutrition, quality, and storage efficiency of fruits and vegetables. During storage of potatoes and strawberries, the VC concentrations in the experimental groups and control groups all decline with time; however, the VC concentrations in the experimental groups are always higher than the control groups (Figure 8). On the eighth day, the VC concentrations in the CTS-TiO$_2$-Ag, CTS-TiO$_2$, and CTS groups of potatoes were 55.67, 57, 62.73 mg/100 g, respectively, which an increase by 13.9%, 5.26%, and 3.0%, respectively, from that of the control group (54 mg/100 g). VC concentrations in strawberries changed in different ways. The VC concentrations in CTS-TiO$_2$-Ag, CTS-TiO$_2$, and CTS groups of strawberries were 14.27%, 9.87%, and 3.71% higher, compared with the control group (Figure 8b). During the whole experimental period, the VC concentrations in film-coated potatoes and strawberries were all higher than in the corresponding control groups, and the VC concentrations in the CTS-TiO$_2$-Ag groups were always the highest, indicating that film coating can effectively inhibit the decline of VC concentrations and enhance the antioxidant capacity of fruits and vegetables, thereby ensuring food quality.

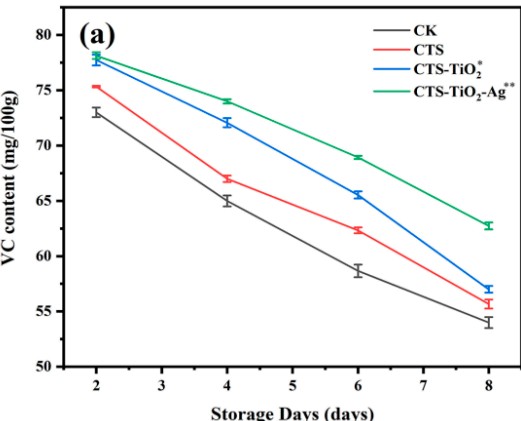
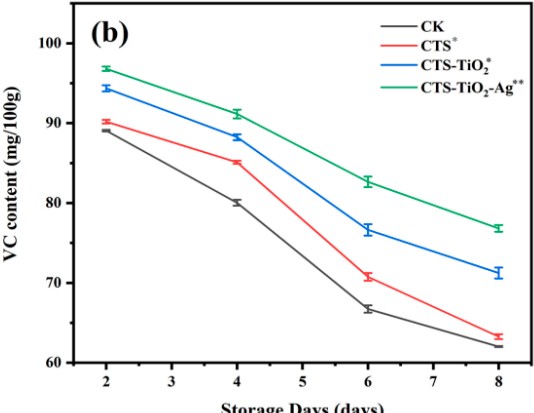

**Figure 8.** Effect of three different materials on VC content after treatment: (**a**) potato; (**b**) strawberry. CK: Potatoes and strawberries without any treatment; CTS: CTS applied to form film-treated potatoes and strawberries; CTS-TiO$_2$: CTS-TiO$_2$ applied to form film-treated potatoes and strawberries; CTS-TiO$_2$-Ag: CTS-TiO$_2$-Ag applied to form film-treated potatoes and strawberries. Data given demonstrate the average of at least three replications (n = 3) ± SD. Asterisks within rows indicate significant differences (* $p < 0.05$, ** $p < 0.01$, *** $p < 0.001$), according to Student's *t*-test.

## 4. Conclusions

(1) Experiments show that an appropriate amount of nano-TiO$_2$ or nano-Ag can improve the physical properties of chitosan. Compared with chitosan films, the light transmittance of nano-Ag-TiO$_2$ and nano-Ag-TiO$_2$-CTS composite films is enhanced, and their bonding strength, aging resistance, antibacterial ability, and water inhibition are effectively strengthened. Such improvements can decrease aerobic respiration of fruits and vegetables, preserve the flavors of fruits, and significantly prolong the fresh periods of potatoes and strawberries.

(2) Comparison of antibacterial spectra among CTS, CTS-TiO$_2$, and CTS-TiO$_2$-Ag films shows that the CTS-TiO$_2$-Ag composite has the broadest antibacterial spectrum, and the addition of nano-TiO$_2$ and nano-Ag significantly enhances the antibacterial ability of films. The inhibition effects of the three materials against *E. coli*, *S. aureus*, and *B. subtilis* were studied. All three materials most significantly resisted *E. coli*, followed by *S. aureus*, however, the inhibitory effects on *B. subtilis* were not significant.

(3) The CTS, CTS-TiO$_2$, and CTS-TiO$_2$-Ag films can all moderately preserve freshness of potatoes and strawberries and can significantly lower the rotting rates and delay the decrease of VC concentrations and weight loss rate of peroxidase activity in fruits and vegetables. In all, the CTS-TiO$_2$-Ag films show the best comprehensive effect. CTS-TiO$_2$-Ag can significantly decrease moisture evaporation, block the entrance of external O$_2$ into films, improve CO$_2$ concentrations in the tissues of fruits and vegetables, and decrease ethylene escape, thereby weakening the respiration and metabolism of fruits and vegetables and achieving freshness preservation.

**Author Contributions:** Conceptualization and methodology, Y.G.; formal analysis and resources, Z.D.; data curation, writing—original draft preparation, R.L.; writing—review and editing, Z.D. and R.L.; Z.D. and R.L. are contributed equally to this work. All authors have read and agreed to the published version of the manuscript.

**Funding:** This research was funded by the College Students′ Innovative Entrepreneurial Training Plan Program of China (202010118001). And the Fundamental Research Funds Project of Modern College of Arts and Sciences, Shanxi Normal University (2019JCYJ13).

**Institutional Review Board Statement:** Not applicable.

**Informed Consent Statement:** Not applicable.

**Data Availability Statement:** Not applicable.

**Conflicts of Interest:** The authors declare no conflict of interest.

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
