# Peer review of "Antibacterial and Freshness-Preserving Mechanisms of Chitosan-Nano-TiO2-Nano-Ag Composite Materials"

_coatings, doi:10.3390/coatings11080914_

Round 1
Reviewer 1 Report
I believe that the presented manuscript is very valuable from a scientific point of view, however, the Authors should work on the technique and redactively refine the text.
My mainly comment on the text:
1.I have a problem with the description of "Antibacterial performances" (line 156): is Oxford cup methods the same as disc diffusion test? If so, then zone sizes in this method are measured from the edge of the disk to the end of the clear zone. However, in the presented photographs (line 275) these zones of growth inhibition are not visible. What do the red circles in these photos mean? The units in which the results of this assay were reported were also not specified. It was also stated (line 173) that the cultures were incubated for 12 h - please explain why it was not the standard time of incubation (24 h) for bacteria.
2. Figure 7 and tables 3 and 4 show the same data. The same is true for Figure 8 and Tables 5 and 6. Duplication of results is unacceptable for any scientific publication. Therefore, please choose one form of presenting these results (both contain statistical analysis, so there should be no problem with that).
3. The article lacks a discussion of the obtained results with other Authors, while the chapter is entitled "Results and discussion" (line 203). Only a few references from the list are cited in this part: 34, 35, 23, 24, 36 and 37. Please complete the part of the article related to the discussion of the results.
4. If a chapter is called "Conclusions" (line 362), it is unlikely that there is room for citing literature (number 26), as was done in line 383. At this point, it is not known whether the Authors are referring to their results or to a cited item. Please rewording of this section of the manuscript.
5. In the "References" section, there is no consistency in writing journal titles - once they are entries with the current abbreviations, while otherwise they are full names (line 397-398 and 400: Scientia Horticulturae instead of Sci. Hortic. or line 408: Procedia Manufacturing instead of Procedia Manuf.). Check the spelling of all of them and write them down in accordance with the applicable citation rules.
6. The notation of reference numbers is not clear: 22, 25, 26 and 28. Are these chapters in the book or another study? I guess there was no title of the study. Please check and, if necessary, correct these provisions.
Author Response
Dear Editors and Reviewers:
Thank you for your letter and for the reviewers' comments concerning our manuscript entitled "Antibacterial and freshness-preserving mechanisms of chitosan-nano-TiO2-nano-Ag composite materials" (coatings-1313174). Those comments are all valuable and very helpful for revising and improving our paper, as well as the important guiding significance to our researches. We have studied comments carefully and have made correction which we hope meet with approval.
Revised portion are marked in red in marked manuscript. The main corrections in the paper and the responds to the reviewer’s comments are as follows, should you have any questions, please contact us without hesitate.
Response to Reviewer 1 Comments
1: I have a problem with the description of "Antibacterial performances" (line 156): is Oxford cup methods the same as disc diffusion test? If so, then zone sizes in this method are measured from the edge of the disk to the end of the clear zone. However, in the presented photographs (line 275) these zones of growth inhibition are not visible. What do the red circles in these photos mean? The units in which the results of this assay were reported were also not specified. It was also stated (line 173) that the cultures were incubated for 12 h - please explain why it was not the standard time of incubation (24 h) for bacteria.
The basic principles of the Oxford cup method and the wafer diffusion method are the same, and both evaluate the drug effect on the tested microorganisms according to the size of the last inhibition zone. Because our synthetic material is liquids, but liquids will flow outside of circular filter paper, we will use the Oxford cup. Although the Oxford cup may not well affect the primary diffusion hair, we can also explain the problem according to the experimental results. In Line 293, in the Oxford cup experiment diagram are (1) CTS; (2) CTS-TiO2; (3) CTS-TiO2-Ag Composite material pair (a) Escherichia coli; (b) Staphylococcus aureus; (c) Antibacterial property of Bacillus subtilis. The figure clearly shows an approximately circular bacteriostatic ring outside the Oxford cup. We measured and statistically analyzed it, and further proved the experimental results by adding Table 1. The red circles in the figure help readers to observe the sizes of the bacteriostatic circles. Clearly, the bacteriostatic circles are shaped in three, and we mark one with the red circle. On line 184, after incubation for 12, 24, and 36 hours with the culture, no significant changes were found in the results, so we recorded all of them, followed by the first 12 hours. We will change from 12 hours to 24 hours.
Changes 1: See Line 307-329 and Table 1.
2: Figure 7 and tables 3 and 4 show the same data. The same is true for Figure 8 and Tables 5 and 6. Duplication of results is unacceptable for any scientific publication. Therefore, please choose one form of presenting these results (both contain statistical analysis, so there should be no problem with that).
Thank you very much for your suggestion. In order to see the change trend more intuitively, we chose the figure, and in the figure with significant analysis, also marked in the legends in Figures 6, 7 and 8.
Changes 2: See lines 332-337, 361-366 and 377-382.
3: The article lacks a discussion of the obtained results with other Authors, while the chapter is entitled "Results and discussion" (line 203). Only a few references from the list are cited in this part: 34, 35, 23, 24, 36 and 37. Please complete the part of the article related to the discussion of the results.
Thank you very much for your suggestion. We have added the results discussed with other authors in the results and discussion section. We added references 38, 39, 42, 43, 45 and 46 to the results and discussion sections.
Changes 3: See lines 221, 227, 291, 329 and 355.
4: If a chapter is called "Conclusions" (line 362), it is unlikely that there is room for citing literature (number 26), as was done in line 383. At this point, it is not known whether the Authors are referring to their results or to a cited item. Please rewording of this section of the manuscript.
Thank you very much for your suggestion, we deleted the cited references in the conclusion.
Changes 4: See lines 400-420.
5: In the "References" section, there is no consistency in writing journal titles - once they are entries with the current abbreviations, while otherwise they are full names (line 397-398 and 400: Scientia Horticulturae instead of Sci. Hortic. or line 408: Procedia Manufacturing instead of Procedia Manuf.). Check the spelling of all of them and write them down in accordance with the applicable citation rules.
Thank you very much for your suggestion. In the reference section, the periodical format of each reference has been changed to abbreviation.
Changes 5: See lines 434-532.
6: The notation of reference numbers is not clear: 22, 25, 26 and 28. Are these chapters in the book or another study? I guess there was no title of the study. Please check and, if necessary, correct these provisions.
References 22, 25 and 26 are the contents of Iop Conference. Reference 28 is from Key Engineering Materials, which specializes in publication of international conference proceedings and monographic collections. We have revised references 22, 25, 26 and 28 according to your suggestions.
Changes 6: See lines 478-490.
We appreciate for Editors/Reviewers’ warm work earnestly, and hope that the correction will meet with approval. Once again, thank you very much for your comments and suggestions.
Reviewer 2 Report
Dear Editor,
Authors developed nanocomposite formulations based on chitosan, nano-TiO2, and nano-Ag and analyzed the antimicrobial activity of these formulations against Escherichia coli, Staphylococcus aureus, and Bacillus subtilis. In addition, the developed formulations were applied for coating strawberry and potato, and some freshness-preserving parameters such as weight loss rate, peroxidase activity, vitamin C content were investigated during storage at room temperature for eight days. My major concern: despite having a section in M&M for statistical analysis of data, this aspect is missing results and discussion. In addition, methods are not described in detail and reference protocols are not appropriately cited to allow others to replicates. Following are some issues that lead to a major revision of the manuscript:
1. Line 32-33: Please give some examples of traditional freshness techniques in this paragraph.
Authors cited several articles in the introduction considering applying chitosan blended with nano-TiO2, and nano-Ag as a coating material with antimicrobial activity. Please, clearly explain the novelty of your experiment compared to the similar studies available in the literature.
Please give the final concentration of nano-TiO2, nano-Ag, chitosan, and glycerol in FFS as wt% or % (w/v)
2. Line 125-126: What is the temperature for dissolving chitosan.
3. Line 129: Please add the information of the volume of FFS cast into the Petri dishes and also the size of Petri dishes.
Please be consistent in writing ml or mL (for example lines 108, 113, 125, 136, 143).
4. Line 142-143: please write the sentence in better words (dissolving nano-TiO2 in 2 gr glycerol is confusing).
5. Line 145-146: Please add the information of the volume of FFS cast into the Petri dishes and also the size of Petri dishes.
For each device (XPS, SEM, spectrophotometer), please add the city and country.
The procedure of doing XPS, SEM, spectrophotometer analysis should be explained briefly. For example, in SEM: the size of the films mounted on the support device, vacuum condition of SEM, the acceleration voltage or in spectrophotometer: The wave range, etc. In addition, the reference protocol for each technique should be cited in the text.
6. Line 158: Please cite the reference protocol for the oxford cup method.
7. Line 160-175: Please rewrite this paragraph in better words and more precisely. For example: What is the final concentration of your bacterial cultures in CFU/mL?
8. Line 184: for how many days samples were analyzed? Please explain the indices briefly or refer them to the corresponding lines.
9. Line 188: “potatoes” should be replaced with “strawberries”.
10. Line 196: Please briefly explain the VC and POD measurements.
11. Line 198: I think the t-test is not a suitable test for the analysis of your results. Please check again with a bioinformatician.
12. Line 203: This section should be started with “3. Results and discussion” not “2”
13. Line 208: Please give the morphology size based on the average length and average diameter. Also, calculate the aspect ratio. This also should be added to M&M and also cite reference protocol. In TEM analysis of the following article clearly show how to calculate this aspect: Characterization of bio-nanocomposite films based on gelatin/polyvinyl alcohol blend reinforced with bacterial cellulose nanowhiskers for food packaging applications. https://doi.org/10.1016/j.foodhyd.2020.106454
14. Line 204: Please give more details in the title. Characterization is so general.
15. Line 225: FT-IR analysis. The successful incorporation of nanomaterial inside the chitosan FFS should be demonstrated by FT-IR. Please add pure chitosan spectrum and also chitosan- nano-TiO2- nano-Ag spectrum. Is the characteristic peak of nanomaterial appeared or not? Also, is it possible to have SEM images from chitosan incorporated with nanomaterial or not?
16. Line 244-252: This paragraph is not discussed properly. You should consider the wavelength between 200-800 nm. In this wave range (200 -800 nm) you have both UV (200-350 nm) and Visible (400-800 nm) and then discuss based on the obtained values in each wavelength.
17. Line 253-263: the transparency of films should be calculated by opacity value or transparency value. The formula also should be added to the M&M: Opacity value or transparency value= -Log transmission at 600 nm/film thickness
Absorption and transparency figures give the same meaning with different graphs. These can be merged into only one image and discussed in one paragraph to avoid repetition.
Antibacterial activity: The diameter of the inhibition zone should be measured and put in a table together with statistical analysis to show which treatment significantly shows higher antibacterial activity against tested bacterial strains.
Selecting the same storage time (8 days) for both strawberry and potato is not clear for me. Generally, potato has much more shelf life than strawberry. Do you have any explanation?
Figure 6: the abbreviation inside the image should be explained in the figure caption. For example: define CK.
Are there any significant differences between the treatments considering weight loss in both potato and strawberry or not? This should be confirmed by doing statistical analysis on the values in table 1 and table 2.
Able 5 and table 6: Please perform the statistical analysis to confirm if there are significant differences between treatments or not.
Author Response
Dear Editors and Reviewers:
Thank you for your letter and for the reviewers' comments concerning our manuscript entitled "Antibacterial and freshness-preserving mechanisms of chitosan-nano-TiO2-nano-Ag composite materials" (coatings-1313174). Those comments are all valuable and very helpful for revising and improving our paper, as well as the important guiding significance to our researches. We have studied comments carefully and have made correction which we hope meet with approval.
Revised portion are marked in red in marked manuscript. The main corrections in the paper and the responds to the reviewer’s comments are as follows, should you have any questions, please contact us without hesitate.
Response to Reviewer 2 Comments
1: Line 32-33: Please give some examples of traditional freshness techniques in this paragraph. Authors cited several articles in the introduction considering applying chitosan blended with nano-TiO2, and nano-Ag as a coating material with antimicrobial activity. Please, clearly explain the novelty of your experiment compared to the similar studies available in the literature. Please give the final concentration of nano-TiO2, nano-Ag, chitosan, and glycerol in FFS as wt% or % (w/v).
Thank you very much for your suggestion. Traditional preservation methods mainly include physical and chemical methods. Physical methods including low-temperature preservation, controlled atmosphere preservation, and electromagnetic radiation, require special equipment, complex operation, and high cost. Chemical methods mainly including spraying fruit and vegetable preservatives will cause health harms, ecoenvironment pollution and other problems.
The existing studies show that compounds synthesized from chitosan, chitosan and salicylaldehyde have antibacterial effects. Nano-TiO2 can kill microorganisms and prolong the storage time of fruits and vegetables. Nano-Ag, nano-Ag-TiO2 composite, and nano-Ag-chitosan composite all have strong antibacterial properties. However, there are few studies on the preservation of fruits and vegetables by using chitosan, nano-TiO2 and nano-Ag composite films. Therefore, it is necessary to comprehensively and deeply study the antibacterial properties and preservation effects of chitosan-nano-TiO2 and nano-Ag.
Nano-TiO2 (wt%) = 1.42%, nano-Ag (wt%) = 0.29%, chitosan (wt%) = 98.33%, and glycerol (wt% )= 0.78% in FFS.
2: Line 125-126: What is the temperature for dissolving chitosan.
Chitosan is dissolved at room temperature.
3: Line 129: Please add the information of the volume of FFS cast into the Petri dishes and also the size of Petri dishes. Please be consistent in writing ml or mL (for example lines 108, 113, 125, 136, 143).
Thank you very much for your suggestion, we have added the manufacturers and specifications of Petri dishes in Main materials and instruments, and unified the units in mL.
Changes 3: See lines 106, 114 and 120.
4: Line 142-143: please write the sentence in better words (dissolving nano-TiO2 in 2 gr glycerol is confusing).
Thank you very much for your suggestion. We have made changes in the original texts. The activated nano-TiO2 (0.035 g) and glycerol (2 g) were added into 100 mL of a glacial acetic acid solution with a volume fraction of 0.6%.
Changes 4: See lines 142-143.
5: Line 145-146: Please add the information of the volume of FFS cast into the Petri dishes and also the size of Petri dishes. For each device (XPS, SEM, spectrophotometer), please add the city and country. The procedure of doing XPS, SEM, spectrophotometer analysis should be explained briefly. For example, in SEM: the size of the films mounted on the support device, vacuum condition of SEM, the acceleration voltage or in spectrophotometer: The wave range, etc. In addition, the reference protocol for each technique should be cited in the text.
Thank you very much for your suggestion. We added the country and model of the instrument to the main materials and instruments, and quoted the reference scheme of each technology. References 30, 31 and 32.
Changes 5: See lines 101-112 and 157-163.
6: Line 158: Please cite the reference protocol for the oxford cup method.
We have quoted the implementation plan of oxford cup. References 33.
Changes 6: See lines 166.
7: Line 160-175: Please rewrite this paragraph in better words and more precisely. For example: What is the final concentration of your bacterial cultures in CFU/mL?
We have modified this paragraph to include concentrations of 107-108 cfu/mL for all three bacterial suspensions tested.
Changes 7: See lines 176-187.
8: Line 184: for how many days samples were analyzed? Please explain the indices briefly or refer them to the corresponding lines.
The experiment was conducted for a total of eight days, during which the indicators were measured every two days for a total of four times.
9: Line 188: “potatoes” should be replaced with “strawberries”.
Thank you very much for your suggestion. Here is our writing error, which has been corrected.
Changes 9: See lines 200.
10: Line 196: Please briefly explain the VC and POD measurements.
Peroxidase (POD) activity was tested by guaiacol colorimetric method. Some samples (1 g) were collected and 5 mL 20 mmol/L KH2PO4 was added in to grind into homogenate in a mortar. The homogenate was then centrifuged for 15 min at the rate of 4000 r/min and supernate was collected and stored at cool places. Residues were extracted once more by 5 mL KH2PO4 solution. The supernate in twice extraction was mixed. Two cuvettes with an optical diameter of 1 cm were used. One was added in with 3 mL reaction mixture and 1 mL KH2PO4 as the zero-calibration control group. The other one was added with 3 mL reaction mixture and 1mL of above enzyme solution. The manual time-keeping was started on immediately OD at 470 nm was measured by a spectrophotometer. The values were read every 1 min and variation of ODs every minute was used to express enzyme activity.
Reaction mixture: 50 mL 100 mmol/L pH=6.0 phosphate buffer and 28 uL guaiacol were mixed by stirring on a magnetic stirring apparatus under heating until the guaiacol was dissolved. After the solution was cooled, 19 uL 300 g/L hydrogen peroxide was mixed in evenly. We also quoted the solutions in the references.
VC: A 1 % HCl solution, a 100 ug/mL VC standard storage solution, and VC standard liquids were prepared. The chopped and mixed fresh samples were weighed (each 1.0 g), ground with acid, and then filtered into 50-mL volumetric bottles at constant volume. Each tested solution (10 mL) was sucked into a test tube. Standard curve plotting: The standard VC solution (0.00, 1.00, 2.00, 4.00, 8.00, 10.00 mL) was sucked into 10-mL colorimetric tubes with acid, and diluted with water into a series of VC standard solutions at concentration of 0, 10, 20, 40, 80, 100 ug/mL respectively, followed by measurement of absorbance. Then a standard curve was drawn with concentration as x-axis and the corresponding absorbance as y-axis. The curve is a straight line passing through the origin, and the slope of the curve can be determined. The absorbance of the samples was measured at 245 mm. We also quoted the solutions in the references.
11: Line 198: I think the t-test is not a suitable test for the analysis of your results. Please check again with a bioinformatician.
When the number of samples is relatively small, Student t-test, which can be divided into single-population test, double-population test and paired sample test, is mainly used to compare whether the difference between two averages is significant. It is more often applied to the confidence test of small-sample judgment.
12: Line 203: This section should be started with “3. Results and discussion” not “2”.
Thank you very much for your suggestion. Here is our writing error, which has been corrected.
Changes 12: See lines 216.
13: Line 208: Please give the morphology size based on the average length and average diameter. Also, calculate the aspect ratio. This also should be added to M&M and also cite reference protocol. In TEM analysis of the following article clearly show how to calculate this aspect: Characterization of bio-nanocomposite films based on gelatin/polyvinyl alcohol blend reinforced with bacterial cellulose nanowhiskers for food packaging applications. https://doi.org/10.1016/j.foodhyd.2020.106454.
We have calculated the average length and width of SEM images, and quoted your recommended references in the results and discussion section. Reference 38.
Changes 13: See lines 221-224.
14: Line 204: Please give more details in the title. Characterization is so general.
Thank you very much for your suggestion. We have revised the title and added details.
Changes 14: See lines 217.
15: Line 225: FT-IR analysis. The successful incorporation of nanomaterial inside the chitosan FFS should be demonstrated by FT-IR. Please add pure chitosan spectrum and also chitosan- nano-TiO2- nano-Ag spectrum. Is the characteristic peak of nanomaterial appeared or not? Also, is it possible to have SEM images from chitosan incorporated with nanomaterial or not?
The infrared comparison between TiO2 and TiO2-Ag uncovered characteristic peaks of Ag ion as well as various internal functional groups, and thus can prove that Ag ion was incorporated. After the chitosan is added, the material as-synthesized is liquid, and can only be extended to form a film under the irradiation of an infrared lamp. The infrared spectrum can only be used for measuring solid and organic liquid. We need an aqueous solution as a solvent, so the liquid material as-synthesized is not affected. As the absorbance of the aqueous solution is large, too little transmitted light can lead to unreliability of infrared spectrum measurement. Therefore, we did not use infrared testing of the liquid-formed films after the addition of chitosan, and we characterized the films using a UV-Vis spectrophotometer.
16: Line 244-252: This paragraph is not discussed properly. You should consider the wavelength between 200-800 nm. In this wave range (200 -800 -nm) you have both UV (200-350 nm) and Visible (400-800 nm) and then discuss based on the obtained values in each wavelength.
Thank you very much for your suggestion. We have modified the discussion section of this paragraph to include comparisons and discussion of different wavelength ranges.
Changes 16: See lines 262-266.
17: Line 253-263: the transparency of films should be calculated by opacity value or transparency value. The formula also should be added to the M&M: Opacity value or transparency value= -Log transmission at 600 nm/film thickness.
Absorption and transparency figures give the same meaning with different graphs. These can be merged into only one image and discussed in one paragraph to avoid repetition.
Antibacterial activity: The diameter of the inhibition zone should be measured and put in a table together with statistical analysis to show which treatment significantly shows higher antibacterial activity against tested bacterial strains.
Selecting the same storage time (8 days) for both strawberry and potato is not clear for me. Generally, potato has much more shelf life than strawberry. Do you have any explanation?
Figure 6: the abbreviation inside the image should be explained in the figure caption. For example: define CK.
Are there any significant differences between the treatments considering weight loss in both potato and strawberry or not? This should be confirmed by doing statistical analysis on the values in table 1 and table 2.
Table 5 and table 6: Please perform the statistical analysis to confirm if there are significant differences between treatments or not.
As for absorption value and transmittance, we illustrate two problems. One is that we synthesized three kinds of films that can absorb light at various bands, and the other is that we synthesized three kinds of films that can transmit light at various bands. Although both of them can explain the light absorption by the films, our separate discussions can further explain that the light-absorbing ability is strong and prove that the light-transmitting ability is weak. We can analyze the effects of different light bands on the three films according to the trend of spectra measured by the ultraviolet spectrophotometer.
We have added the statistical table and analysis of bacteriostatic circle diameter, and further explained which treatment showed higher bacteriostatic effect on which bacteria, as shown in Table 1.
Our way was to cut a bevel, and then on the bevel daub, we synthesized three kinds of films. Because there was no protection of cortex, storage time will also be shortened. Since strawberry has thin skin and easily rots, we directly put the three kinds of films on the surface of strawberry. Hence, we used the same time and strawberry processing, and the results are enough to illustrate the problems.
Thank you very much for your suggestion. We have written the representatives of each line in Figures 6, 7 and 8 in the legends.
We have already expressed the significant results in the legends through the statistical analysis of the tables. The first reviewer thought our figures and tables were duplicate and asked us to choose the intuitive figures as the main contents. We have written the significant results in the legends of the figures, and all the data are significant. We have added significant analysis to the graphs and legends of Figures 6 and 8. Data given demonstrate the average of at least three replications (n = 10) ± SD. Asterisks within rows indicate significant differences (*p < 0.05, **p < 0.01, ***p < 0.001), according to Student’s t-test.
Changes 17: See lines and 307-329, 333-337, 362-366, 378-382 and Table 1.
We appreciate for Editors/Reviewers’ warm work earnestly, and hope that the correction will meet with approval. Once again, thank you very much for your comments and suggestions.
Round 2
Reviewer 1 Report
I have read the explanations and comments from the Authors, which I consider sufficient for the publication of this manuscript.
Reviewer 2 Report
All my comments have been addressed.
This manuscript is a resubmission of an earlier submission. The following is a list of the peer review reports and author responses from that submission.